# Research Progress on the Improvement of Flame Retardancy, Hydrophobicity, and Antibacterial Properties of Wood Surfaces

**DOI:** 10.3390/polym15040951

**Published:** 2023-02-15

**Authors:** Hao Jian, Yuqing Liang, Chao Deng, Junxian Xu, Yang Liu, Junyou Shi, Mingyu Wen, Hee-Jun Park

**Affiliations:** 1Department of Wood Material Science and Engineering Key Laboratory, College of Materials Science and Engineering, Beihua University, Jilin 132013, China; 2Department of Housing Environmental Design, Research Institute of Human Ecology, College of Human Ecology, Jeonbuk National University, Jeonju-si 54896, Republic of Korea

**Keywords:** wood surface, flame-retardant, hydrophobic, antibacterial

## Abstract

Wood-based materials are multifunctional green and environmentally friendly natural construction materials, and are widely used in decorative building materials. For this reason, a lot of research has been carried out to develop new and innovative wood surface improvements and make wood more appealing through features such as fire-retardancy, hydrophobicity, and antibacterial properties. To improve the performance of wood, more and more attention is being paid to the functioning of the surface. Understanding and mastering technology to improve the surface functionality of wood opens up new possibilities for developing multifunctional and high-performance materials. Examples of these techniques are ion crosslinking modification and coating modification. Researchers have been trying to make wooden surfaces more practical for the past century. This study has gradually gained popularity in the field of wood material science over the last 10 years. This paper provides an experimental reference for research on wood surface functionalization and summarizes the most current advancements in hydrophobic, antibacterial, and flame-retardant research on wood surfaces.

## 1. Introduction

Wood is a raw material that is renewable and has good mechanical properties [1], as well as being inexpensive and having abundant reserves. It has been extensively employed in furniture, construction, and other industries. In addition to being a material for furniture and interior decoration, it can have applications in high-tech sectors [2,3,4,5,6,7]. Thousands of years ago, the Romans and Egyptians used alum and vinegar to reduce the combustibility of wood [8]. For the most part, lignin, cellulose, and hemicellulose make up the xylem of wood [9] (Figure 1). It is prone to thermal disintegration at medium and high temperatures, which releases a lot of heat, smoke, and hazardous fumes, starting fires and posing a major risk to human life [8,9,10]. The majority of researchers use added flame retardants [11] to make wood fire-resistant using straightforward physical techniques. Most researchers tend to apply additive flame retardants to make wood fire-resistant through simple physical methods. The most common types are intumescent flame retardants. The application range of traditional carbon sources is significantly constrained by their high hygroscopicity and propensity to precipitate on matrix surfaces during processing. Additionally, as petrochemical resources are consumed more, traditional carbon sources—mostly those derived from petroleum-cracking products—become more expensive and non-renewable. Biomass replaces traditional carbon sources with high carbon content, such as chitosan [12], cellulose [13], lignin [14], starch [15], etc. are often applied as char-forming agents. The traditional acid source promotes the dehydration of the char-forming agent into carbon, and the traditional melamine is applied as the gas source. Three fundamental components make up intumescent flame retardants.

In the xylem, hydroxyl functional groups are found in cellulose, hemicellulose, and lignin, with -OH displaying the greatest hydrophilicity [16,17,18,19]. Wood is a common material applied in building, furniture, shipbuilding, and other sectors because it is organic, renewable, biodegradable, low-carbon, and environmentally beneficial [20,21,22,23,24]. However, because of its hydrophilicity, wood is easily cracked, molded, and degraded, which significantly reduces its useful life in the industry, limits its widespread application in industrial production and wastes resources [25,26,27]. Numerous investigations have demonstrated that the solution to this issue lies in the alteration of the hydrophilic hydroxyl groups on the surface of the wood. The majority of investigations typically employ metal ions to cross-link hydroxyl groups; however, wood surfaces can also be made hydrophobic using modified adhesives. On the one hand, the process of metal–ion crosslinking modification is straightforward, simple to apply, quick to react, has the low temperature required and has a high crosslinking density. On the other hand, the substrate’s mechanical characteristics can be improved by changing the coating. The coating can effectively withstand the expansion and cracking deformation of the base due to its high tensile strength and strong elongation. The hydrophobicity of wood surfaces can be determined by measuring the contact angle (WCA) and rolling angle (WRA) between water and wood surfaces.

Generally, wood is less dense than metal. There are many holes on the surface and inside of wood. It is easily contaminated by various liquids and propagated bacteria [28,29]. On the one hand, bacteria and fungi can degrade and destroy cellulose, hemicellulose, and lignin in the wood xylem, resulting in wood discoloration, mildew, and warping, greatly reducing the mechanical properties and service life of wood [30,31]. On the other hand, bacteria, fungi, viruses, parasites, etc. also pose a major threat to human health, and wood is mainly applied in commercial spaces such as homes, offices, hotels, and restaurants. Wood, the most often applied raw material, is readily a breeding ground for bacteria on surfaces such as floors, tables, counters, cutlery, etc. [32] is generally impossible to remove with standard cleaning techniques. The introduction of hazardous microbes into wood products applied for certain locations or applications ought to be minimized [33]. Wood’s xylem structure can be weakened by fungi, which also lessens the wood’s mechanical qualities. For instance, some researchers have applied the selective degradation of wood cell wall components by the white-rot fungus as a delignification technique. The lignin is modified to produce a “sticky” lignin radical by white-rot fungal enzyme systems including peroxidase and laccase-mediator systems [34,35]. A simple and commonly applied method to improve the fireproof, waterproof, and antibacterial properties of wood and wood products is to spray functional coatings [36]. Gamini P. Mendis et al. [37] modified the epoxy resin covering to make it fire-resistant, which had a clear fire retardant impact on the wood surface. In addition, a layer-by-layer self-assembly technique [38] is used to form a fireproof, hydrophobic, or antibacterial layer [39] on the wood surface. By alternately depositing a substrate in a polyelectrolyte solution with an opposing charge, layer-by-layer self-assembly creates a multilayer polyelectrolyte film. Layer-by-layer self-assembly method was applied by Fu et al. [40] to generate a composite coating of chitosan, phytic acid, and TiO_2_-CuO on a wood surface at 60 degrees Celsius.

The majority of plants in the natural world, including wood, are made of polymers. These polymers are employed as primary components in the creation of production equipment and everyday objects. Additionally, epoxy resin, polyurethane, and other coatings can be created chemically from polymer ingredients to offer wood surfaces functionalization using the coating modification approach.

Following the COVID-19 outbreak, the demand for timber has been on the rise and is anticipated to do so in the upcoming years. The development trend for wood composite materials in the future is to enhance the surface function of wood and its flame-retardant, hydrophobic, and antibacterial qualities [10]. Then, the three sections of this essay are flame-retardant, hydrophobic, and antibacterial, respectively. The discussion of various modification techniques was applied. To advance the study of functional wood, the modification techniques for functionalization of wood in recent years are compiled.

## 2. Improvement of Flame Retardancy of the Wood Surface

Due to the flammability of wood and its widespread application in everyday life, fire has become a common hidden threat. Flame retardants have a wide range of applications [41,42]. People’s deep understanding of the carbon peak and the concept of carbon-neutral, halogen-free, and friendly low-carbon flame retardants has been paid increased attention [43,44,45]. When the intumescent flame-retardant (IFR) [46,47,48] is heated, the carbonizing catalyst causes the carbonizing agent to dehydrate into carbon, and the carbides form a carbon layer with a fluffy porous closed structure under the influence of the gas produced when the intumescent agent is broken down. Once created, it is not flammable in and of itself and reduces heat transfer from the heat source to the wood, which stops the gas from spreading. The wood will self-extinguish if there is insufficient oxygen and fuel for burning [49,50,51,52,53,54]. Adding intumescent biomass flame-retardant is one of the easiest, most economical, and most effective ways to improve the flame retardancy of wood surfaces [41].

Chitosan is a non-toxic, biodegradable natural biopolymer [55], and is the second largest natural fiber after cellulose [56]. Chitosan contains amino groups and hydroxyl groups in its molecular structure, which can be applied as a carbon source for intumescent flame-retardant systems and is widely applied in the field of wood flame-retardant research [57]. Cho Whirang et al. [58] applied chitosan and nitromethylenephosphonic acid (NTMP) ionically cross-linked to produce fire-resistant wood coatings at 50 °C. Figure 2 shows the crosslinking reaction. The vertical burning test of wood samples treated with the modified coating reaches the V-0 level. When the wood is heated to 800 °C, the residual carbon content reaches 29%, and a highly flame-retardant foamed char is formed on the surface of the wood. The THR and pHRR both fell by 99.3% and 96.7%, while the thermal decomposition temperature of wood decreased by roughly 50 °C. The findings demonstrate that the combination of P and N elements results in a P-N condensed carbon layer, which has a good flame-retardant effect on the surface of the wood.

One technique for making wood fireproof is spraying. Some of the coatings used to modify flame-retardant wood materials by spray application are epoxy, acrylic and polyurethane resins. Epoxy resin offers the strongest adhesion to wood surfaces of the bunch. Epoxy [59] is a coating with exceptional mechanical, thermal, and chemical resistance. Although it is also commonly used in daily life, some applications are severely hampered by its intrinsic flammability. To modify the wood to be flame-retardant, many researchers employ epoxy resin as an adhesive and spray it on the wood’s surface. To create an epoxy-based (WBEP) sintered fire protection coating based on biomass water, Li et al. [60] first created a biomass flame-retardant using chitosan (CS), ammonium polyphosphate-coated melamine-formaldehyde resin (MFAPP), and organic montmorillonite (OMMT). A vertical burning test, LOI test, TGA test, SEM analysis, CONE analysis, Raman spectroscopy, FTIR Fourier transform infrared spectroscopy, and other techniques were applied to investigate the flame-retardant impact of synthetic biomass flame-retardant coatings on wood surfaces. The outcomes indicate that the coating with an 18% flame-retardant component has the best flame-retardant performance. With an LOI of 31.8%, wood samples passed the vertical burn test with a UL-94 V-0 rating. The thermal degradation initiation temperature fell from 278 °C to 254 °C, and the residual carbon reached 23.9% at 800 °C, according to the TGA experiment data. After combustion, the carbonization layer was large, dense, and complete, according to scanning electron microscopy (SEM). The -OH group in CS combines with NH^4+^ in APP to produce H_2_O and NH_3_, which dilutes combustible gases such as O_2_ according to FTIR analysis. To create the long-bonded carbon-carbon layers P=O, P-O-P, and P-O-C, carbonize CS and EP. The I_D_/I_G_ value changed from 3.99 to 2.83, the degree of graphitization of the carbon layer increased, and the intensity of combustion decreased with the addition of CS/MFAPP/OMMT, according to the Raman spectroscopy data. The total heat release (THR) of the WBEP-18%/CS/MFAPP/OMMT-coated wood composite dropped from 43.6 MJ/m^2^ to 26.6 MJ/m^2^, resulting in strong carbon layer strength and thermal stability, according to the cone calorimetry data.

Huang et al. [61] applied the P-Cl bond of hexachlorocyclotriphosphazene and the -OH of vanillin to generate hexa-(4-aldehyde-2-methoxy-phenoxy)-cyclotriphosphazene (HCPV) by the nucleophilic substitution reaction, and then oxidized the aldehyde group of HCPV to carboxyl group through Pinniker oxidation reaction. Hexa-(4-carboxyl-2-methoxy-phenoxy)-cyclotriphosphazene (HCPVC) was prepared, and the HCPVC-cured epoxy resin was coated on the surface of the wood. The response time at room temperature was 36 h. The mechanism diagram is shown in Figure 3. After the vertical combustion test, the sample reached the V-0 level, and the LOI value also reached 30.7%. The experiment showed that the modified wood surface flame-retardant performance was excellent. At the same time, the Fourier transform infrared spectroscopy and cone calorimeter test showed that the P-O-C bond in the HCPVC-EP coating was more easily decomposed than the C-C bond, which promoted the formation of an expanded and dense phosphate-rich carbon layer on the wood surface, and the residual carbon amount reached 37.1% at 700 °C. The formation of the char residue layer effectively inhibited the transfer of heat and prevented the thermal degradation of the wood materials. It had a more efficient function of protection against fire.

After reacting eugenol with ethyl acetate at 5 °C for 24 h, Zhong et al. [62] produced the intermediate chemical PPDEG, and then applied a Prilezhaev epoxidation reaction to create a novel epoxy flame-retardant coating. The synthesis route is shown in Figure 4. As the stretching vibration peak of C=C at 3527 cm^−1^ and the stretching vibration peak of -OH at 1464 cm^−1^ disappeared, the characteristic absorption peak of the epoxy group appeared at 931 cm^−1^, and the new epoxy flame-retardant coating (PPDEG-EP) was successfully synthesized. The eugenol-modified flame-retardant coating significantly improved the flame-retardant property of the wood surface, and UL-94 reached V-0, LOI increased from 21% before treatment to 32.1%, the heat release rate (HRR) of wood improved by coating was significantly reduced, the total heat release rate (THR) was 15.1% lower than that of untreated wood, and the residual carbon rate reached 35.16% at 800 °C. It was proved that the coating has excellent flame retardancy.

Some scholars believed that industrial adhesives were difficult to process and potentially toxic to human health and the environment. Therefore, they were more inclined to use layer-by-layer (LbL) self-assembly technology with good flame-retardant performance. It was environmentally friendly and could well preserve the natural properties of wood [63]. Dong et al. [64] first assembled a polyethyleneimine (PEI) -ammonium polyphosphate (APP) polyelectrolyte composite coating on the wood surface with layer-by-layer self-assembly (LbL) technology to give the wood surface high fire resistance and smoke suppression ability. The crosslinking effect is shown in Figure 5. Then, the flame-retardant on the wood surface was further enhanced with metal ions (Cu^2+^, Co^2+^) crosslinking. The increase in P and N elements detected by SEM-EDX indicates the successful application of PEI-APP coating. The increase in Cu and Co elements shows the successful crosslinking of metal ions and coating, and the surface of the wood has a smooth waxy structure. The chemical bond composition of the coating was characterized by Fourier transform infrared spectroscopy (FT-IR), and the coating applied to the wood surface was identified as a PEI-APP coating with bivalent metal–ion crosslinking. Thermogravimetric analysis (TGA) showed that the T_max_ of (PEI-APP)_15_-Cu^2+^/Co^2+^ coated wood was nearly 35 °C lower than that of untreated wood, and the residual carbon content increased from 21.8% to 42.8% and 42.3% at 800 °C. Metal ions (Cu^2+^, Co^2+^) can not only act as a catalyst to promote ammonium polyphosphate to produce NH_3_ and reduce the concentration of flammable gas but also act as a crosslinking agent to form a dense carbon layer on the wood surface and enhance its flame retardance. The limit oxygen index (LOIs) increased from 23.5% to 47% and 42.5% after the Cu^2+^ and Co^2+^ crosslinking treatment of PEI-APP coating on the wood surface. A cone calorimetry test showed that compared with untreated wood, Cu^2+^, and Co^2+^ crosslinking treated PEI-APP coated wood reduced the total smoke emission by 16.7% and 9.6%, the average heat emission rate by 32.6% and 18.7%, and the total heat emission by 20.7% and 19.5%, respectively.

There are several advantages to the coating modification methods. However, most of the raw materials used in modified coatings are not environmentally sound, which is the subject of current research. One of the issues in the future will be the research of biomass-based materials such as chitosan.

Layered double hydroxides (LDHs) [65] have the advantages of non-pollution, low smoke, and non-toxicity, and a small amount added has little impact on the properties of wood. Due to the water and carbon dioxide between layers and its solid phase flame-retardant effect, it can significantly reduce the heat release rate (HRR) of wood and is a friendly and efficient halogen-free flame-retardant [66,67]. Guo et al. [68] first created a layer of intermediates using the dipping approach before reacting at 100 °C for 8–10 h in a stainless steel autoclave lined with Teflon. At 60 degrees Celsius, the wood forms to create a coating of magnesium-aluminum hydrotalcite. Compared with uncoated wood, the LOI value increased from 18.9% to 39.1%, HRR decreased from 283 kW/m^2^ to 145 kW/m^2^, and THR also decreased by 40%. The results showed that the Mg-Al hydrotalcite coating had a very efficient effect on improving the flame retardancy of the wood surface. Hu et al. [69] synthesized calcium–aluminum hydrotalcite with Root Cutting Silicate Layers (CaAl-SiO_3_-LDHs) using calcium nitrate, aluminum nitrate, sodium silicate, and sodium hydroxide as raw materials. The experiment showed that the heat release rate of wood treated with hydrotalcite was significantly reduced, the ignition time was extended by 14 s, the flame retardancy of CaAl-SiO_3_-LDHs was excellent, and the fire resistance of the wood surface was significantly improved.

In the application of various flame retardants, the sol–gel method has gathered great interest from either the academic and industrial communities. It is expected that sol–gel chemistry will still play a key role in flame retardance in the next years with its great potential being further exploited [70]. Li et al. [71] applied the sol–gel method to mineralize silica in wood pores and cell cavities and applied silica to prepare mineralized wood composites with flame-retardant properties. The results of CONE analysis showed that the ignition time (TTI) of mineralized wood is 21 s. The peak heat release rate associated with cone calorimetry decreased by 45.8%.

The study on the flame resistance of wood surfaces is compiled in this section. Despite the flaws of some of the experimental methodologies, hopefully it will enlighten readers. Furthermore, chitosan was employed as a raw material in the Cho Whirang experiment, and the results showed a considerable flame-retardant impact, demonstrating that environmentally friendly biomass raw materials will eventually replace conventional flame-retardant polymer raw materials.

## 3. Improvement of Hydrophobicity of the Wood Surface

Wood is a popular material for furniture and floor manufacturing as well as other industries since it is inexpensive, simple to produce, beautifully patterned, naturally renewable, and low-carbon [44,72,73,74,75]. However, wood cell walls are composed of cellulose, etc., and the hydroxyl groups on its branch chains combine with water to create hydrogen bonds, resulting in the wood itself being significantly hydrophilic [76,77]. It is prone to corrosion in a humid environment, resulting in cracking, mildew, greatly reducing the durability of wood, resulting in resource waste, and economic losses [78,79,80]. The improvement of wood’s surface hydrophobic property can effectively improve its water-resistance, form an anti-fouling and anti-mildew surface, reduce unnecessary damage, and prolong its service [72,81].

One of the most commonly used methods for hydrophobic modification is metal–ion coordination. It is mainly carried out on the hydrophilic groups present on the surface of the wood. Song et al. [82] applied metal ions Zr^4+^, Fe^3+^, Al^3+^, Ni^2+^, Co^2+,^ and Zn^+^ to coordinate the hydroxyl group on the wood surface to make a hydrophobic wood surface, among which the surface modified by Zr^4+^ had the best hydrophobic effect. Figure 6 displays the diagram of the mechanism. The experimental results showed that the surface water contact angle perpendicular to the wood development direction was 145° and the surface water contact angle along the wood growth direction was 139° under a drying temperature of 80 °C after dipping in Zr^4+^ solution of 0.5–0.75 wt.% for the 30 s. The hydrophobicity of the wood surface was greatly increased, as were the surface’s wear resistance and ability to self-clean.

Wang et al. [83] only applied a non-toxic chemical solution of silver nitrate (AgNO_3_) during the whole fabrication process. First, wood substrates were used to deposit Cu thin films. Then, it was immersed in an AgNO_3_ solution with a concentration of 2 mM to form an Ag-Cu-wood surface. Finally, after a reaction time of 5 min, a superhydrophobic coating was obtained on the wood surface by heating at 100 °C for 1 h. According to the findings, the Ag-Cu-wood surface’s WCA was close to 160.5° and its WSA was almost 0°. Ag-Cu-wood retained high hydrophobic performance despite abrasion and scrape testing, and it has promising application possibilities in the construction and civil engineering industries.

In addition to metal ions, researchers used organic solvents to coordinate hydroxyl groups to make the wood surface hydrophobic. Tang et al. [84] directly dipped wood in a tetramethyl cyclo tetrasiloxane D4H modification solution to obtain highly hydrophobic wood. The preparation mechanism of the hydrophobic wood surface was that the hydrophobic methyl (-CH_3_) group is grafted onto the wood surface to finally obtain hydrophobized wood by the dehydrogenation reaction of the -OH groups on the wood surface with -Si-H bonds on the D4H structure. Figure 7 depicts the process. The experimental findings demonstrated that the transverse and longitudinal portions of the treated wood displayed good hydrophobicity at a temperature and humidity of 25 °C and 60%, respectively. The modified wood’s wear resistance and self-cleaning qualities were successfully tested at the same time, offering a straightforward and user-friendly technique for increasing the hydrophobicity of wood surfaces.

The self-adhesive properties of polydopamine [85] are favored by researchers. One of the techniques with a high rate of application for hydrophobizing wood surfaces is the application of a polydopamine hydrophobic coating. Using layer-by-layer self-assembly technique, Shao et al. [86] created a hydrophobic coating of polydopamine (PDA) cross-linked 1H,1H,2H,2H-perfluoro-decyl trichlorosilane (PFDTS) on the surface of wood. By using polydopamine’s self-adhesion, they reduced the roughness of the wood’s surface. The results of the experiments demonstrated that a superhydrophobicity coating was successfully formed on the wood surface with PFDTS = 2 vol% and PDA≥2 mg/mL, and the WCA between the wood surface and water surpassed 150°and RA was between 4.8° and 4.9°. Duan et al. [87] applied polydopamine, Cu nanoparticles, and fluoro silane to create a superhydrophobic coating on wood surfaces and tested its hydrophobicity under several special environmental conditions. The experimental findings demonstrated that different liquids on the wood surface exhibited excellent hydrophobic properties when the concentration of Cu^2+^ solution was 50 mM, WCA > 150°, and SA < 10°. Hydrophilic pollutants rolled down with the water droplets, the self-cleaning ability was significantly improved, and the wear resistance of the superhydrophobic coating was also significantly improved.

Yang et al. [88] believed that the introduction of chemical bonds was an important way to improve the superhydrophobic durability of a wood surface. The strong viscosity of polydopamine was applied as the bottom layer, and nano-Al_2_O_3_ was used to prepare a micro-nano rough structure. Then 3-mercaptopro-pyltriethoxysilan (KH580) was used to connect the polydopamine and alumina. Finally, fluorine-free octadecyl trichlorosilane (OTS) was used to make a low-functionalized surface. It provided a new strategy for the preparation of a superhydrophobic wood surface. The one-step and two-step methods are shown in Figure 8 and Figure 9, respectively. The results showed that the CA and SA of Wood@One-step were 155.2° and 4.76°, and those of Wood@Two-step were 152.9° and 7.65°, respectively. In both the one-step and two-step methods, the CA and SA on the wood surface were all greater than 150° and less than 10°, which significantly improves the hydrophobic effect and has good development prospects.

The sol–gel method is a common method for obtaining a hydrophobic wood surface among wood surface modification methods. This will be the next direction of research in the future. In the study of Sanja Kostic et al. [89]. The TEOS sol–gel was applied to obtain a hydrophobization of the wood surface. The combination of both layers (TEOS + APTES) resulted in a higher contact angle, which was around 100°.

Tsvetkova et al. [90] developed a “sol–gel@paint” composition based on wax and silicone. The laboratory tests were performed on the pine wood coated with developed coatings. Due to the addition of colloidal silica with a hydrophobized surface (aerosol) to the “sol–gel@paint” compositions, the contact angle θ reached 110°. The coatings based on a mixture of wax and TEOS, as well as the ones with the addition of silicone block copolymer practically reached the level of super-hydrophobicity (θ = 145–152°). Laboratory studies of the developed coatings revealed that the coating based on the wax-TEOS-derived “sol–gel@paint” composition has the greatest effect on hydrophobicity.

In this section, the hydrophobic alteration primarily begins with the hydrophobic group -OH. Crosslinking modification and coating modification are the two principal applications. Both approaches have benefits and drawbacks.

## 4. Improvement of Antibacterial Properties of the Wood Surface

Wood is a common building material since it is an organic substance that is renewable. It will erode by fungi as a result of the temperature and humidity in the environment, leading to mildew and degradation [91]. Additionally, since wood is employed in the food business, health and safety concerns have naturally taken precedence. Growing attention has been shown in coating wood surfaces with antibacterial technologies [92,93].

Numerous studies have shown that a variety of dangerous microorganisms are presented on wood surfaces that people frequently come into contact with, especially furniture. In addition, because of the numerous pores in the structure of wood, bacteria can easily penetrate the surface of the wood and lurk there. This can have a significant impact on human health. Widespread interest has been shown in research on antimicrobial capabilities [93]. Gaurav Sharma et al. [94] made an antibacterial modification of the wood surface with the grafting method. Acrylonitrile (AN) and ethyl acrylate (EA) were grafted on the wood surface by changing monomer and initiator concentrations, temperature, time, pH value, etc., to make an antibacterial wood surface. The experimental results showed that the growth of E. coli on pine was inhibited (AN/EA). The largest inhibition zone was observed on an 8 mm or so strip, and the diameter of the inhibition zone was 21 mm.

Zhang et al. [95] applied a solvothermal method where nanoparticles (TiO_2_ and Ag) were chemically bonded to the wood surface through a combination of hydrogen groups. The R% of Ag-doped TiO_2_/wood (ATW) reached 99.0% and 90.5% toward E. coli and S. aureus, respectively. The researchers believed that tetravalent Ti ions in TiO_2_ would be substituted by monovalent Ag ions, resulting in oxygen vacancies, which might be a benefit for the strengthening of the antibacterial properties of the wood surface.

Tran-Ly et al. [96] applied an antibacterial coating with fungal melanin and vegetable oil as a compound to prevent the corrosion of wood by bacteria and destroy the precious value of some wood products. The research results showed that plant essential oils had strong antibacterial effects, especially linseed oil and tea tree oil, and the antibacterial rate of the longitudinal section of wood was lower than 1%, even after two weeks, and the survival rate for streptococcus was still less than 10%.

Angelika Macior et al. [97] functionalized wood with poly(methylmethacrylate) (PMMA) and poly(2-(dimethylamino)ethyl methacrylate) (PDMAEMA) to yield wood grafted with PMMA-b-PDMAEMA-Br copolymers with antibacterial properties. The synthetic route is shown in Figure 10. Antibacterial tests revealed that no bacterial growth was observed in places with direct contact between the wood blocks (Figure 9) and S. aureus and E. coli inoculum. It had good antibacterial properties.

Wood’s natural ability to absorb water makes it simple to foster the growth of fungus, which not only deteriorate wood and shorten its useful life but also release spores into the environment that pollute the environment and damage human health [98,99]. Qi et al. [100] prepared Nano-AgCu Alloy to treat wood surfaces. It had an antibacterial effect on Aspergillus niger, Penicillium citrinum, and Trichoderma viride. The diagram is shown in Figure 11. According to the findings, the antibacterial rate was above 75% and the leaching rate was just 7.678% when the concentration reached 1000 mg/L.

Dai et al. [101] applied water-based nano silver to carry out antibacterial modification treatment on the surface of the wood and characterized its antibacterial properties. The experimental findings demonstrated that when the retention rate of silver reached 0.324 g·m^−2^, the antibacterial rates of the wood surface for three kinds of molds, namely Aspergillus niger V. Tiegh, Penicillium citrinum Thom, and Trichoderma viride Pers. ex Fr, reached 80%, 75%, and 80%, respectively. The leaching rate of nanosilver is only 4.75%.

Bi et al. [102] prepared modified solvents with chitosan and cinnamaldehyde as raw materials. After treating wood with the modified solvents, they studied the inhibitory effect of cinnamaldehyde chitosan modified solvent on the growth of Aspergillus niger on the wood surface. The findings indicate that when the molar ratio of -CHO to -NH_2_ in the modified solvent was 3:1, the antibacterial rate of the wood surface was the highest, at 95.8%. The results of the experiment demonstrated that the cinnamaldehyde chitosan solution increased the stability of cinnamaldehyde and significantly boosted the antibacterial rate of the wood surface. 

Lazim et al. [103] prepared a starch-based anti-fungal coating with buttercup that greatly improved the antibacterial type of wood surface without changing other good properties of wood as much as possible. The characterization and analysis were carried out by TG, Fourier transform infrared spectroscopy (FTIR), and scanning electron microscopy (SEM). The coating was evenly distributed on the surface of the wood, and the morphology was slightly rough. The antibacterial experiment showed that the mass loss rate of the uncoated wood samples reached 60% after 120 days of white-rot bacteria erosion, while the mass loss rate of the wood samples coated with antibacterial coating was only 10%. The coating had an obvious inhibition effect on white-rot bacteria.

The inhibition of bacteria and fungi is discussed in this section. Among them, the method of silver loading experiment has a very noticeable antibacterial effect on wood surface, which may be an indication of the course of future antibacterial experiments.

## 5. Conclusions

The application of wood has grown along with human society, and increased emphasis is being dedicated to studying the surface qualities of wood that make it hydrophobic, antibacterial, and fire-resistant. By ion crosslinking and coating modification, the properties of flame-retardant, hydrophobic, and antibacterial were applied to wood with corresponding success.

The three primary lines in this study are flame-retardant, hydrophobic, and antibacterial, while the branch lines are crosslinking, coating, and other modification techniques. We found that the wood modification procedure occasionally produces some pollutants, although being effective in functionalization. The search for extremely functional wood materials is the only factor that gave rise to these techniques.

The major approach to enhancing the surface function of wood has increasingly shifted toward low-carbon, halogen-free, and non-toxic alteration techniques as a result of ongoing contemplation on human development and environmental issues. Future development will focus heavily on finishing with hydrophobic, antimicrobial, and flame-retardant properties using biomass as a raw material.

It is evident that numerous further alteration strategies are being investigated. A lot of work is also being done using sol–gel techniques that were tested on wood (but not exclusively for modified wood). They are seen to be potentially significant ways to alter the course of wood’s future. Enhancing the functionality of wood is still a hot topic in wood research in general. The researchers improved their approach and the data they compiled in the publication are available for research and reference. In the future, it is anticipated that new and better techniques will be developed to enhance the flame retardancy, hydrophobicity, and antibacterial properties of wood surfaces.

## Figures and Tables

**Figure 1 polymers-15-00951-f001:**
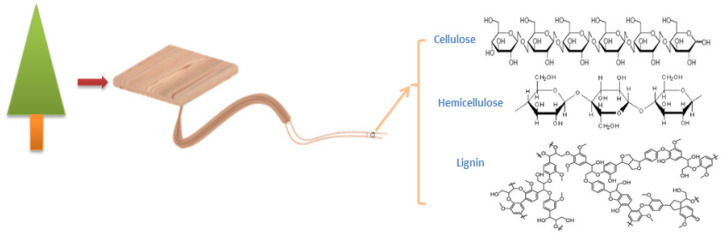
Xylem structure.

**Figure 2 polymers-15-00951-f002:**
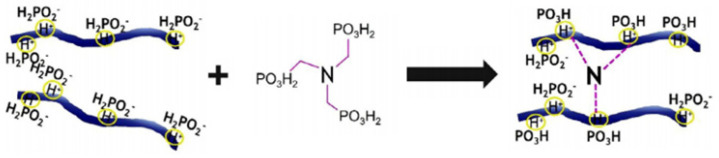
Schematic illustration of crosslinking reaction between chitosan and nitrilotris (methylenephosphonic acid) (NTMP) [58].

**Figure 3 polymers-15-00951-f003:**
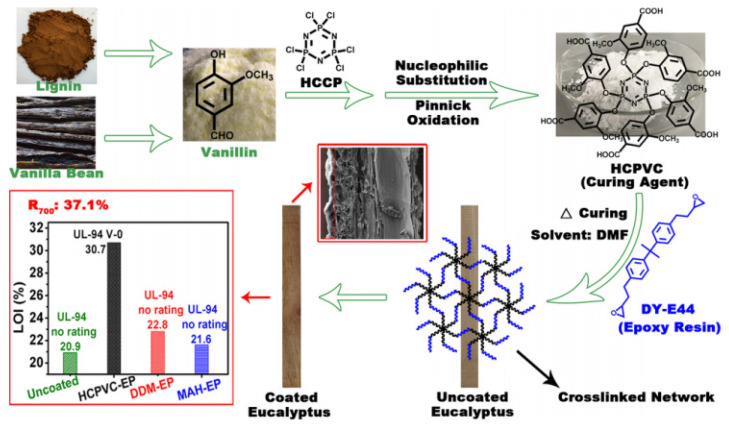
Mechanism diagram of HCPVC curing epoxy resin on the wood surface [61].

**Figure 4 polymers-15-00951-f004:**
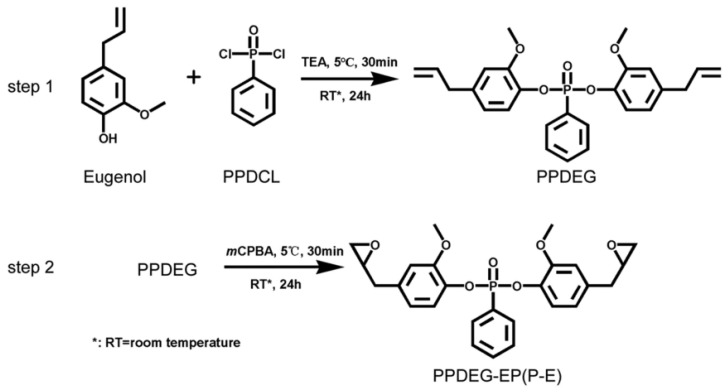
Two-step synthesis route of PPDEG-EP [62].

**Figure 5 polymers-15-00951-f005:**
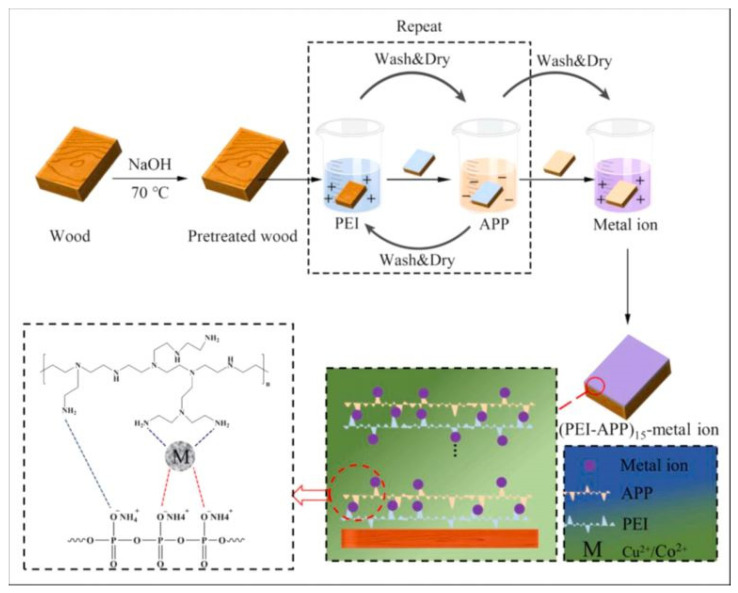
Schematic diagram of flame-retardant coating construction and metal–ion crosslinking effect [64].

**Figure 6 polymers-15-00951-f006:**
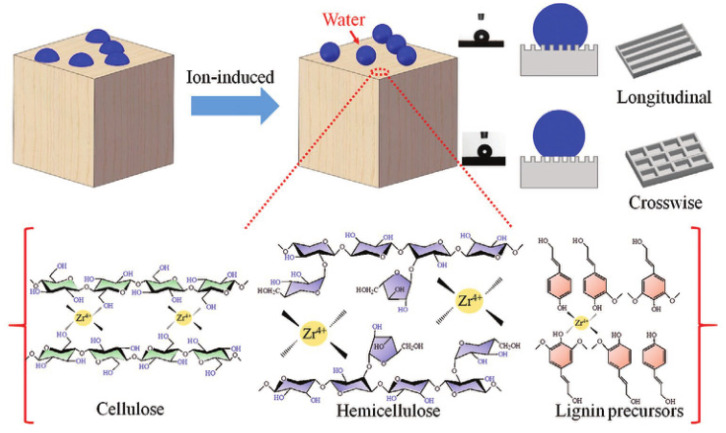
Schematic illustration of metal–ion induced wettability transition over wood surface [82].

**Figure 7 polymers-15-00951-f007:**
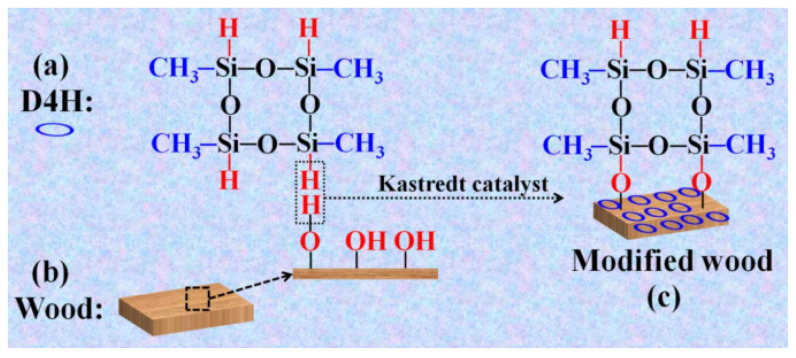
Schematic representation of the process for obtaining the D4H modified wood [84].

**Figure 8 polymers-15-00951-f008:**
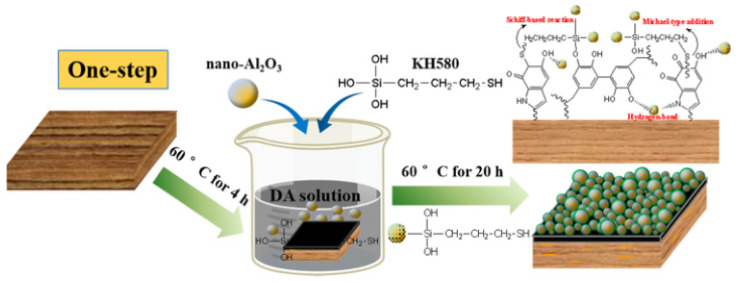
Schematic diagram of one-step [88].

**Figure 9 polymers-15-00951-f009:**
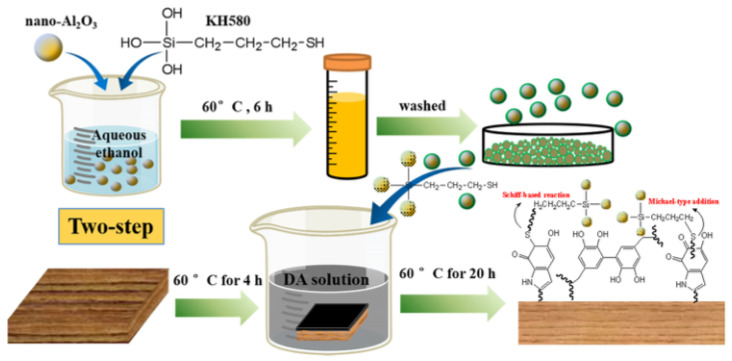
Schematic diagram of two-step [88].

**Figure 10 polymers-15-00951-f010:**
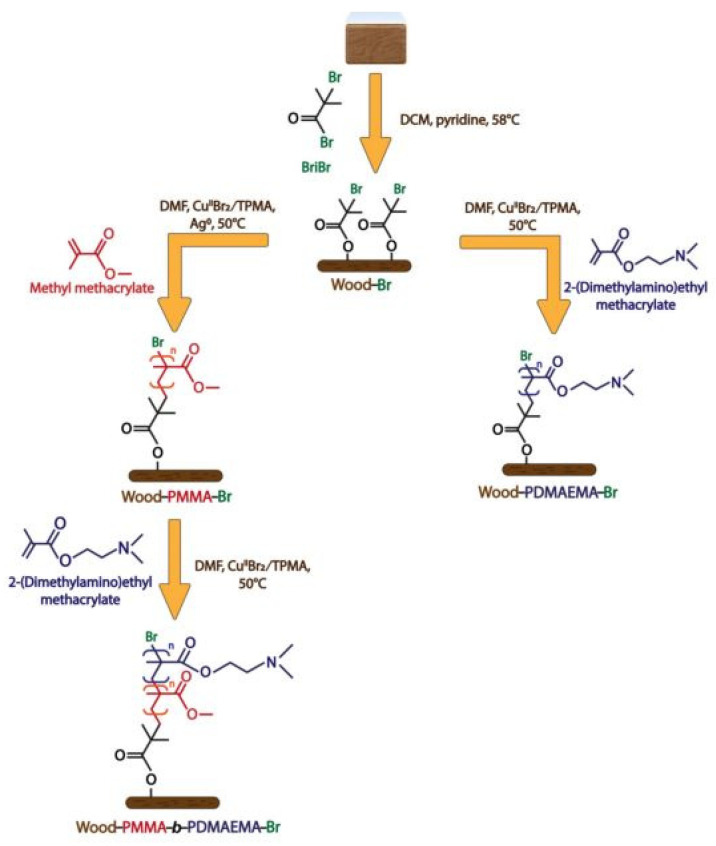
A two-step synthetic route for the preparation of a wood–polymer composite [97].

**Figure 11 polymers-15-00951-f011:**
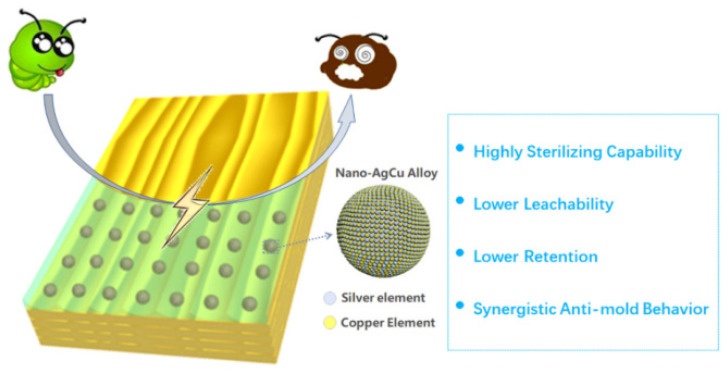
Schematic illustration of the nano silver-copper alloy (nano-AgCu) for mold resistance on the surface of wood [100].

## Data Availability

Not applicable.

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
