# Peer review of "Research Progress on the Improvement of Flame Retardancy, Hydrophobicity, and Antibacterial Properties of Wood Surfaces"

_polymers, 2023, doi:10.3390/polym15040951_

Round 1

Reviewer 1 Report

This review paper is about the improvement of the wood surface with the fire-retardant, hydrophobic, and antibacterial properties through the use of the ion cross-linking and coating modification methods

The review summarizes the fire-retardant, antibacterial, etc. modifications of wood. The novelty of the article is pretty low. The review of the article tried to avoid as much as possible the broad topics in the review related to all of the wood modification techniques since there are many published articles (2018-2022) about coating and chemical wood modification techniques and I did not find any type of critical review.

The introduction wasn’t so good. Some parts should be explained more and compared clearly with the traditional methods and modern approaches. In the traditional methods, these design strategies all have one thing in common is that the surface hydroxyl groups need to be blocked or destroyed but with crosslinks, hydrophilic groups (−OH) through metal ions increases the surface roughness and alters the microstructures that will also contribute to the surface hydrophobicity. They should explain more about the advantages of cross-linking modification and coating modification in the other parts.

Not all of the experimental protocols are given in sufficient detail.  The concentration or reaction times are absent. In the improvement of flame retardancy of the wood surface part, these coating systems are not eco-friendly methods.

Chitosan-based formulations – intended to be used as flame retardant coatings - were prepared by ionic crosslinking of hypophosphorous acid-modified chitosan with nitrilotris(methylenephosphonic acid) (NTMP).

To improve the curing and flame retardancy of epoxy resin, P-Cl bond of hexachlorocyclotriphosphazene and the -OH of vanillin to generate hexa-(4-aldehyde-2-methoxy-phenoxy)-cyclotriphosphazene (HCPV) by the nucleophilic substitution reaction, and then oxidized the aldehyde group of HCPV to carboxyl group through Pinniker oxidation reaction was illustrated in Fig 3.

The other one is new type of epoxy flame retardant coating by nucleophilic substitution reaction and Prilezhaev epoxidation used eugenol and benzoyl chloride (PPDCL) in two steps in Fig 4.

Apart from the coating system, they emphasize mostly on the crosslinking mechanism. The flame retardant on the wood surface was further enhanced with metal ions (Cu2+, Co2+ ) cross-linking. The increase in P and N elements indicates the successful application of PEI-APP coating. The increase in Cu and Co elements shows the successful cross-linking of metal ions and coating, and the surface of the wood has reduced the smoke emission. The experimental methods were shown in Fig.5.

The article should convince the reader about the superiority of cross-linking modification and coating modification relative to other well-known chemical modification methods in terms of synthesis, applications, properties, cost, and weight. There is only a literature survey of the published articles about 2 types of wood modification techniques.

The discussion should focus on specific properties of wood surfaces or address specific structural problems in wood modifications and the ways to overcome these challenges. Around 65-70% of the current review content discusses the applications of wood composites which are not needed since there are many reviews already discussing this research gap.

The summary in the conclusion part needs to be improved. They can suggest some methods or solutions that kind of new and more perfect methods to improve the flame retardancy, hydrophobicity and antibacterial properties of wooden surfaces. For future work, hydrophobic woods with considerable potential can be suggested for application in oil−water separation or pollutant treatment with low-cost and scalable.

Some important information are missing in the review. The article should also mention the type of wood species because modification on the different wood types also varies due to different roughness, pore type, and amount of lignin, cellulose, or hemicellulose in the different wood species that should be explained in the text. Additionally, the second modification technique includes the improvement of the hydrophobicity of the wood surface. With other chemical modification techniques like antibacterial properties, the surface should also be hydrophobic. I think the review article should be categorized into 1 part which are hydrophobicity. Then they can discuss the topics like the application of hydrophobic wood surface modification for flame retardancy and antibacterial applications. Also, for flame retardancy modification techniques, there is no information about the as positive effect of the surface hydrophobicity on the flame retardancy properties. They didn’t mention the environmental impact, sustainability, and industrial-scale production of these coatings.

1.      Line 13-16: The sentence is too long and hard to read.

2.      Line 28: Reference 1 is about cellulose nanofibers. Choosing an article whose main topic is wood will be more suitable.

3.      Line 28-29: The sentence is incomplete.

4.      Line 32: What is xylem?

5.      Line 38: What are intumescent flame retardants?

6.      Line 40: Why are they looking for an alternative to traditional carbon sources? give the cons of them.

7.      Line 45-47: This general information about wood should be placed in the first part of the introduction. Repeating it is not necessary.

8.      Line 50: incomplete sentence.

9.      Line 53: Mentioning surface roughness is irrelevant here.

10.   Line 67-69: repeating expressions

11.   Line 71-75: Little information about methods.

12.   Line 76-79: be striking. Before part 2, introduce the chapters with a summary. It will increase the flow.

13.   Line 85-87: this sentence could be rewritten more scientifically.

14.   Line 96: there are same phrases.

15.   Line 109: Combine spray and epoxy before detailing the epoxy.

Overall, I think the manuscript does not meet the acceptance criteria.

Reviewer 2 Report

This review paper probably can be considered for publication in Polymers. However, before consideration, some points should be revised, namely:

1)      The English is poor, so the meaning of some sentences is lost. Please improve English through the paper.

2)      Introduction. The statement “Since the density of wood is lower than that of metal, its surface and interior have many holes…” is false.  Please rewrite to be correct.

3)      Introduction. Relation of the paper to the polymer science should be highlighted and justified in detail.

4)      Line 154. Do you mean Figure 4? Please correct.

5)      Sections 2 – 4. Please give the expanded summaries for every sections.

6)      Conclusions should be expanded to be more sound.

Reviewer 3 Report

The paper addresses an important issue related to the protective layers of wood. In particular, non-flammability, antibacterial. In my opinion, the review conducted is fine, although it contains minor shortcomings. For example, the authors omitted the broader context of the use of the sol-gel method. They dedicate little space to this important method. It is noteworthy that, for such an extensive number of works cited, they comment relatively little on the results cited. It would be worth expanding on this, especially since this is a review paper. In summary, in my opinion, the work meets the criteria and as such can be considered for publication, but after absolutely expanding the subject matter to include the SOL-GEL method. 

Round 2

Reviewer 1 Report

Authors' response is sufficient

Reviewer 2 Report

Thank you for revision, now the paper can be accepted

Reviewer 3 Report

The paper is improved and can be accepted